# Australian vegetated coastal ecosystems as global hotspots for climate change mitigation

Oscar Serrano (iD) et al.[#]

Policies aiming to preserve vegetated coastal ecosystems (VCE; tidal marshes, mangroves and seagrasses) to mitigate greenhouse gas emissions require national assessments of blue carbon resources. Here, we present organic carbon (C) storage in VCE across Australian climate regions and estimate potential annual $CO_2$ emission benefits of VCE conservation and restoration. Australia contributes 5–11% of the C stored in VCE globally (70–185 Tg C in aboveground biomass, and 1,055–1,540 Tg C in the upper 1 m of soils). Potential $CO_2$ emissions from current VCE losses are estimated at 2.1–3.1 Tg $CO_2$-e $yr^{-1}$, increasing annual $CO_2$ emissions from land use change in Australia by 12–21%. This assessment, the most comprehensive for any nation to-date, demonstrates the potential of conservation and restoration of VCE to underpin national policy development for reducing greenhouse gas emissions.

D estruction and degradation of natural ecosystems accounts for 12–20% of the $CO_2$ released into the atmosphere globally[1]. Despite their relatively small global extent (between 0.5 and $1 \times 10^6$ km², equivalent to 0.2% of the ocean surface), vegetated coastal ecosystems (VCE), tidal marshes, mangroves and seagrasses, contribute ~50% of C sequestered in marine sediments[2] (i.e., blue carbon), with their organic carbon (C) sequestration rates exceeding those of terrestrial forests, per unit area, by 1–2 orders of magnitude[3]. Hence, conservation and restoration of VCE has an important potential to contribute to climate change mitigation[4,5].

Blue carbon ecosystems are among the most threatened ecosystems on Earth. The global area occupied by VCE is being globally reduced at rates ranging from 0.03 to >1% per year, twice as high as those reported for tropical forests[3,5]. These losses led to the development of blue carbon strategies to prevent and mitigate greenhouse gas (GHG) emissions through the conservation and restoration of VCE. The development of programs like REDD+, the payments for ecosystem services[6] and the inclusion of VCE within Nationally Appropriate Mitigation Actions[7] aim to maintain the benefits these ecosystems provide to climate change mitigation and adaptation, fisheries, and other ecosystem services that support coastal communities and their livelihoods[5,8]. Blue carbon strategies are now being included within Nationally *Determined* Contribution to mitigate and adapt to climate change. However, this requires strong scientific evidence, and whereas reports of C stocks and sequestration rates in VCE have recently increased exponentially[5,9–11], comprehensive estimates of blue carbon storage at national and continental scales are lacking, particularly for tidal marshes and seagrass. Uncertainties on the extent of these ecosystems, their C stocks and sequestration rates, as well as limited available data on the loss and fate of C after disturbance, hinder adoption of VCE into carbon trading and national inventories[5–7].

Here, we pioneer the estimation of C stocks in aboveground biomass and soils, as well as soil C sequestration rates, in VCE at the national level, and do so for the Australian continent, one of the major reservoirs of VCE in the planet. We estimate the potential for VCE conservation and restoration to mitigate GHG emissions in Australia and demonstrate, therefore, the potential of blue carbon strategies to support policies contributing to climate change mitigation at the national level.

## Results

**Australian blue carbon**. Total C stocks in aboveground biomass and the upper 1-m of VCE soils in Australia were 67–183 Tg C and 1053–1542 Tg C, respectively, with annual soil C sequestration rates of 3.5–5.5 Tg C year⁻¹ (Fig. 1, Table 1).

The extent, geographic distribution and type of VCE determine the distribution of C stocks and sequestration rates over the continent (Fig. 1). Mangroves contain ~80% of total C in living aboveground biomass of Australian VCE, while seagrasses accounted for ~70% of total soil C stocks and sequestration rates. Most C stocks in seagrass and tidal marsh ecosystems are found in their soils (98% and 99%, respectively), while C stocks in mangrove ecosystems are distributed in both soil (62%) and aboveground biomass (38%) pools (Table 1). Australian mangroves have up to 17-fold and 65-fold higher C stocks in aboveground biomass per unit area compared to tidal marshes and seagrasses, respectively ($P < 0.001$; Fig. 1), while mangrove soil C stocks and sequestration rates are 2-times and 3-times higher than tidal marshes and seagrasses, respectively ($P < 0.001$; Fig. 1, Supplementary Table 1). The soil C stocks per unit area in tidal marshes are 1.5-fold higher than in seagrass meadows ($P < 0.01$; Fig. 1). These values are consistent with global estimates,

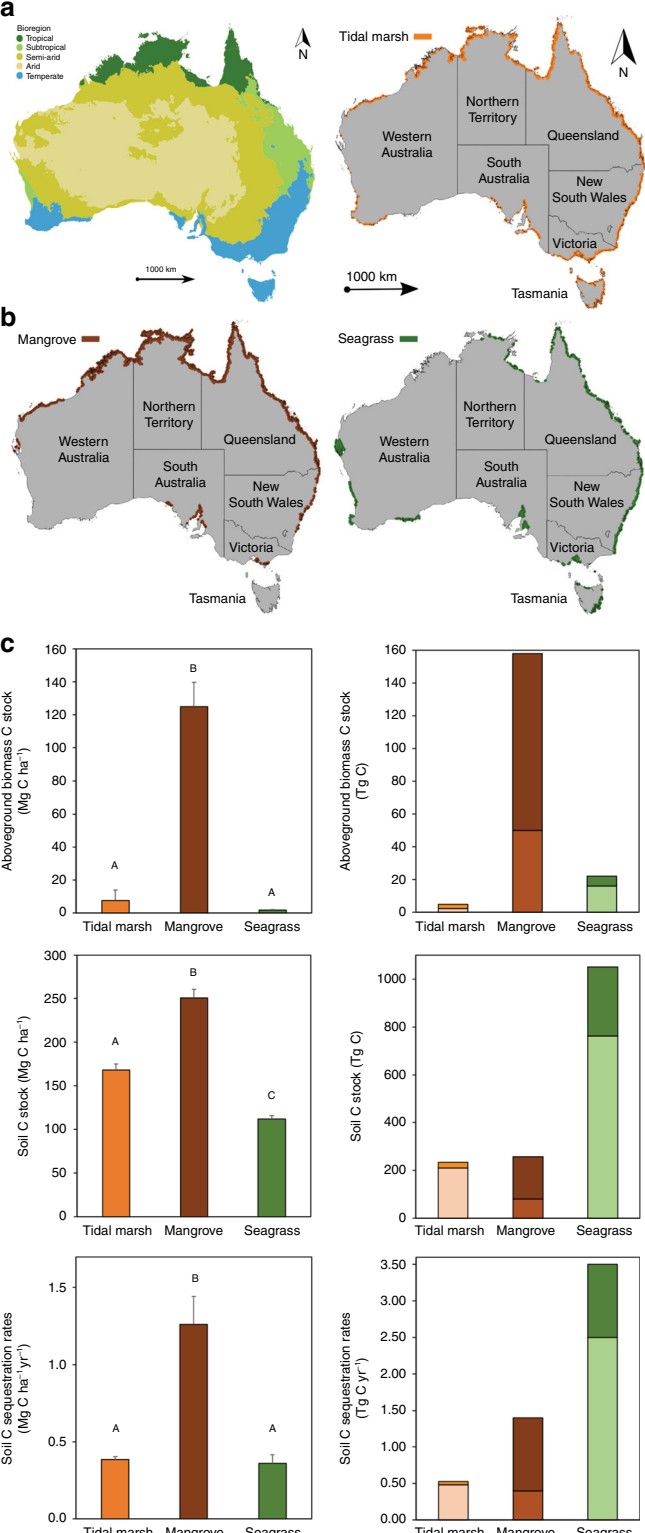

with higher C storage capacity of mangroves and tidal marshes compared to seagrasses on an areal basis[3,5].

Most tidal marsh and mangrove ecosystems in Australia occur in tropical regions (62% and 73%, respectively), while seagrasses are distributed across subtropical (38%), tropical (32%), and arid (16%) climate regions (Supplementary Table 2). Tidal marsh and mangrove soil C stocks and sequestration rates are 3-fold to 13-fold higher in tropical Australia than in other climate regions, mainly due to their extensive coverage over broad intertidal

**Fig. 1**. Distribution of climate regions, vegetated coastal ecosystems (tidal marshes, mangroves and seagrasses) and organic carbon (C) storage in Australia. **a** Climate regions used to classify vegetated coastal ecosystems and scale blue carbon storage across Australia. Climate regions for Australia were modified from the Australian Bureau of Meteorology's "Koppen–Major Classess" climate classification for Australia based on temperature/humidity, vegetation and seasonal rainfall[61]. The original climate classification scheme was comprised of six classes: Equatorial, Tropical, Subtropical, Desert, Grassland and Temperate, but the number of climate regions was reduced into five categories: Tropical (includes Equatorial), Subtropical, Arid (instead of Desert), Semi-arid (instead of Grassland) and Temperate. **b** Spatial distribution of tidal marsh[25], mangrove[26], and seagrass[28] ecosystems within Australia. **c** Organic carbon stocks in living aboveground biomass and soils (in the top meter), and C sequestration rates per unit area (Mg C ha$^{-1}$) and across Australia (Tg C). The stacked bars represent the maximum and minimum estimates (s.d.). Source data are provided as a Source Data file

saltflats in tropical regions (Supplementary Table 2). Similarly, mangrove soil C stocks and sequestration rates are up to 60-fold higher in tropical compared to other climate regions, owing to their larger extent in the tropics[12]. Subtropical seagrasses within Australia hold 2-fold to 6-fold higher C stocks than seagrasses from other Australian climate regions. However, knowledge of Australian seagrass extent is incomplete due to challenges in mapping this ecosystem, as recently illustrated by the recent discovery of 35,000 km$^2$ of tropical seagrass in the intensively studied Great Barrier Reef[13]. Similarly, the spatial extent of tropical tidal marshes (including high intertidal saltflats) is likely large, but poorly mapped. Hence, the extent of seagrasses and tidal marshes may be significantly larger than currently estimated, so their blue carbon contribution estimated here is a conservative one.

Combined, the Australian VCE soil C stocks and sequestration rates per unit area are up to 3-fold higher in tropical regions compared to other regions ($P < 0.001$; Fig. 2, Supplementary Table 3), while C stocks in aboveground biomass are significantly higher in both subtropical and tropical regions ($P < 0.05$; Supplementary Table 1). The C stocks and sequestration rates per unit area also differ, though not consistently, among the three ecosystem types (Fig. 2, Supplementary Table 3). For tidal marshes, the C stocks in aboveground biomass per unit area are up to 6-fold higher in in temperate regions compared to semi-arid and subtropical regions ($P < 0.05$), while soil C stocks and sequestration rates per unit area are not significantly different among climate regions ($P > 0.05$; Supplementary Table 1). This likely reflects the influence of higher biomass species (e.g., *Juncus* spp. rushes) in many temperate marshes, compared to the lower biomass species (e.g., *Sporobolus virginicus* and *Sarcocornia quinqueflora*) typically found in tropical, arid and subtropical climates. Variability in tidal marsh soil accretion rates and C stocks is often associated with differences in the position within the intertidal zone[14], porewater salinity, sediment inputs, and plant productivity[15].

Tropical mangroves contain up to 2-fold higher C stocks in aboveground biomass per unit area compared to temperate mangroves ($P < 0.001$), and soil C stocks and sequestration rates are up to 2-fold higher in subtropical mangroves compared to other climate regions ($P < 0.05$; Fig. 2, Supplementary Tables 1 and 3), which is in agreement with previous studies[16,17].

**Table 1 Organic carbon (C) storage in Australian vegetated coastal ecosystems (i.e., tidal marshes, mangroves and seagrasses), per unit area (in Mg C ha$^{-1}$ and Mg C ha$^{-1}$ year$^{-1}$) and Australia-wide (in Tg C)**

**a**

| Ecosystem | Stock-aboveground biomass per unit area (Mg C ha$^{-1}$) | | | | Total area (Mha) | | Stock-aboveground biomass (Tg C) | |
|---|---|---|---|---|---|---|---|---|
| | N plots | Mean | Median | SD | Min | Max | Min | Max |
| Tidal marsh | 52 | 7.5 | 6.4 | 6.1 | 1.4 | 1.5 | 2.3 | 2.6 |
| Mangrove | 37 | 125 | 94 | 90 | 0.3 | 1.1 | 50 | 158 |
| Seagrass | 52 | 1.9 | 1.5 | 2.0 | 9.3 | 12.8 | 16 | 22 |
| Total | 141 | | | | 11.0 | 15.4 | 67 | 183 |

**b**

| Ecosystem | Stock-soil (Mg C ha$^{-1}$ in 1 m-thick) | | | | Total area (Mha) | | Stock-soil (Tg C) | |
|---|---|---|---|---|---|---|---|---|
| | N cores | Mean | Median | SD | Min | Max | Min | Max |
| Tidal marsh | 292 | 168 | 140 | 127 | 1.4 | 1.5 | 210 | 234 |
| Mangrove | 262 | 251 | 238 | 155 | 0.3 | 1.1 | 81 | 257 |
| Seagrass | 549 | 112 | 85 | 88 | 9.3 | 12.8 | 762 | 1051 |
| Total | 1103 | | | | 11.0 | 15.4 | 1053 | 1542 |

**c**

| Ecosystem | Sequestration rates-Soil (Mg C ha$^{-1}$ year$^{-1}$) | | | | Total area (Mha) | | Sequestration rates (Tg C year$^{-1}$) | |
|---|---|---|---|---|---|---|---|---|
| | N cores | Mean | Median | SD | Min | Max | Min | Max |
| Tidal marsh | 292 | 0.39 | 0.3 | 0.3 | 1.4 | 1.5 | 0.48 | 0.54 |
| Mangrove | 24 | 1.26 | 0.9 | 0.9 | 0.3 | 1.1 | 0.4 | 1.4 |
| Seagrass | 36 | 0.36 | 0.3 | 0.3 | 9.3 | 12.8 | 2.5 | 3.5 |
| Total | 352 | | | | 11.0 | 15.4 | 3.5 | 5.5 |

Mean and median ± SD C stock in (a) living aboveground biomass and (b) in the top meter of soil
c: Soil C sequestration rates. Mha = 10$^6$ ha

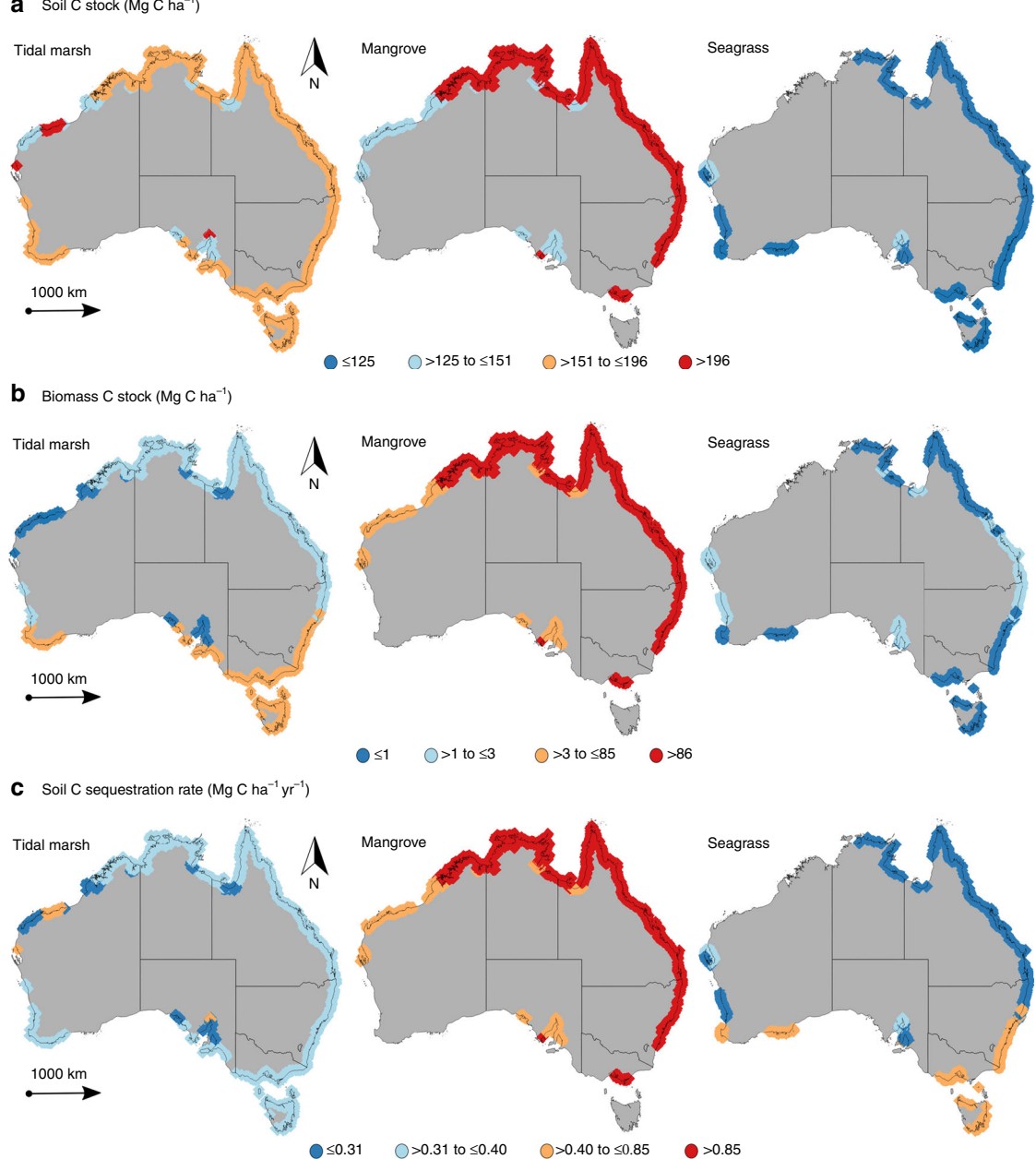

**Fig. 2** Scaled up estimates of organic carbon (C) storage in vegetated coastal ecosystems (tidal marshes, mangroves and seagrasses) across Australian climate regions. **a** Soil C storage (Mg C ha$^{-1}$) in the top meter. **b** Living aboveground biomass C stock (Mg C ha$^{-1}$). **c** Soil C sequestration rates (Mg C ha$^{-1}$ year$^{-1}$). The four ranges of data (indicated by different colors) are based on the lower quartile, median quartile, and upper quartile. Source data are provided as a Source Data file

Subtropical and temperate mangroves in Australia are mainly comprised of a single species, *Avicennia marina*, while tropical Australian mangroves comprise more than 40 species[18]. The species richness, together with climatic and environmental drivers associated with variations in coastal geomorphology (i.e., temperature, rainfall, tides, sediments and nutrients) govern the structure and function of mangroves and tidal marshes, and thereby their C storage capacity, with relatively high plant productivity and biomass in tropical, subtropical and some temperate regions compared to arid regions[9,19,20-22].

Seagrasses inhabiting arid climate regions have 5-fold to 9-fold higher C stocks in aboveground biomass per unit area than those in other climate regions ($P < 0.001$; Fig. 2, Supplementary Table 3). Seagrasses in arid regions also exhibit the highest soil C stocks per

unit area among climate regions ($P < 0.001$), while tropical seagrasses have the lowest soil C stocks ($P < 0.001$). However, seagrass soil C sequestration rates are similar among climate regions ($P > 0.05$), suggesting higher remineralization rates of soil C in tropical regions[4]. The relatively low C stocks of tropical seagrasses may be due to the predominance of small, ephemeral and fast-growing species in the tropics (e.g., *Halophila* spp. and *Halodule* spp.), which also support intensive grazing by sea turtles and dugongs, compared to the persistent meadows with large biomass found in arid and temperate regions (e.g., *Posidonia* spp. and *Amphibolis* spp.), where herbivory rates are much lower[10,23]. Meadows formed by these larger temperate seagrass species contain relatively high amounts of degradation-resistant organic compounds in their tissues (e.g., lignin and cellulose) compared

to small tropical seagrass species, which have readily decomposable tissues[24]. In addition, the scarcity of riverine inputs in arid regions likely results in higher irradiance reaching the seafloor compared to other climate regions with higher run-off, thereby enhancing aboveground and belowground seagrass productivity[13,23] and soil C stocks in arid regions.

Biomass and soil C stocks and sequestration rates for VCE vary among administrative jurisdictions (Supplementary Table 4). Queensland, Northern Territory and Western Australia hold the largest area of tidal marshes (39%, 28%, and 25%, respectively) and mangroves (39%, 37%, and 20%, respectively) within Australia, while Queensland and Western Australia have the vast majority of seagrass ecosystems (65% and 20%, respectively). This information is an essential foundation for the implementation of blue carbon accounting processes and climate change mitigation strategies (i.e., C policies, abatement and trading schemes) in Australia.

**Australia as a global blue carbon hotspot.** Australia is among the countries that hold the largest area of VCE (11–15 Mha, 9–32% of VCE worldwide), including 3–37% of global tidal marshes (1.4–1.5 Mha[3,11,25]), ~2–8% of mangroves (0.3–1.1 Mha[3,17,25,26]) and ~15–43% of global seagrass meadows (9.3–12.8 Mha[5,13,25,27,28]; Table 2). Accordingly, Australia has some of the world's largest blue carbon storage capacity, with 5–11% of global blue carbon soil stocks and 2–7% of annual soil C sequestration, based on available global estimates[3–5,10,11,17,29,30] (Table 2). Soil C stocks per unit area in Australian VCE are similar to global estimates, but their C sequestration rates per unit area are lower than global mean estimates[3,5], particularly for tidal marshes and seagrasses (Table 2). This difference can be partially explained by the relative sea-level stability of Australian coastlines compared to other regions (i.e., North America and Europe) that have experienced rapid sea-level rise over the last millennia, hence, enhanced soil C accumulation[31]. In addition, differences in carbon accumulation rates among climate regions may arise from differences in methods used for estimating soil accumulation rates (i.e., surface elevation tables, $^{210}$Pb, $^{239+240}$Pu, and $^{14}$C) and the periods of accumulation encompassed by the estimates[32]. Nevertheless, the extent of VCE in Australia makes it a global blue carbon hotspot, and the loss of VCE extent since European arrival provides unique opportunities for Australia to mitigate emissions through blue carbon strategies. Indeed, Australia, along with Indonesia, Malaysia, USA and Brazil rank among the nations with greatest potential to benefit from developing blue carbon schemes[4,9,33]. Yet, Australia is, through the estimates provided here, the only nation with a robust estimate of existing blue carbon resources at the national scale.

Soil C stocks in Australian terrestrial ecosystems have been estimated at 29.7 Mg C ha$^{-1}$ (in 30 cm-thick soils[34]), while Australian VCE contain on average ~4-fold higher C stocks (123 Mg C ha$^{-1}$ in 1 m-thick soils) than in terrestrial counterparts. Note that this estimate is conservative since terrestrial and VCE soil C stocks can reach thicknesses >1 m and >6 m depth, respectively[10,12,34]. Furthermore, VCE can accrete vertically over millennia without becoming saturated in C, and soil C stocks are generally protected from fires. As a result, C turnover rates in VCE are an order of magnitude slower than most terrestrial soils[5,35]. Our estimate of the soil $CO_2$ sequestration capacity of Australian VCE (13–20 Tg $CO_2$-e year$^{-1}$) is equivalent to ~4–6% of the $CO_2$ emissions from fossil-fuel burning, cement production and gas flaring in Australia (361 Tg $CO_2$-e year$^{-1}$ at 2014 rates[36]). In addition, the $CO_2$ sequestration capacity of Australian VCE equates to 70–140% of $CO_2$ emissions from land use change in Australia (estimated in 14.4–18.4 Tg $CO_2$-e year$^{-1}$ [37,38]). Hence, VCE are a significant component of Australia's C budget and provide effective opportunities for C sequestration and climate change mitigation strategies based on VCE conservation and restoration.

**Potential of Australian VCE for greenhouse gas mitigation.** Historic losses of VCE extent after European settlement in the 19th century in Australia have been estimated at 13,800 km$^2$ for tidal marsh[14] (47–50% loss of original extent), 11,500 km$^2$ for mangroves[33,39] (52–78% loss of original extent) and 32,000 km$^2$ for seagrass[40] (20–26% loss of original extent). In addition to the loss of important ecosystems services (e.g., coastal protection, fisheries and biodiversity[5]), losses of Australian VCE result in a loss of their $CO_2$ sequestration potential, remineralization of C stocks in aboveground biomass as $CO_2$, and the risk that erosion and remineralization of soil C accumulated over millennia contributes to GHG emissions[4]. Management activities, which fall outside business-as-usual scenarios, aiming to restore VCE can enhance soil C sequestration and/or avoid GHG emissions following disturbance, as demonstrated for Australian seagrass meadows[41], and thus have the potential to be eligible for C crediting under financial mechanisms. The Australian Government Emission Reduction Fund is a policy mechanism that

**Table 2 Extent, soil organic carbon (C) sequestration rates and stocks (in the top meter of soil) in vegetated coastal ecosystems**

| a | Global extension (km²) | | Global soil C sequestration rate (Tg C year⁻¹) | | Global C stock in soil (Pg C) | |
|---|---|---|---|---|---|---|
| **Ecosystem** | **Min** | **Max** | **Min** | **Max** | **Min** | **Max** |
| Tidal marsh | 41,657[11] | 400,000[3] | 4.8[3] | 87.3[3] | 0.67[4,11] | 6.5[5] |
| Mangrove | 137,760[3] | 166,000[17] | 23[29] | 25[29] | 5.0[30] | 6.4[17] |
| Seagrass | 300,000[5] | 600,000[5] | 48[3] | 112[3] | 4.2[10] | 8.4[10] |
| Total | 479,417 | 1,166,000 | 75 | 224 | 14.2 | 21.3 |

| b | Australian extension (km²) | | Australian soil C sequestration rate (Tg C year⁻¹) | | Australian C stock in soil (Tg C) | |
|---|---|---|---|---|---|---|
| **Ecosystem** | **Min** | **Max** | **Min** | **Max** | **Min** | **Max** |
| Tidal marsh | 13,765 (3%) | 15,329 (37%) | 0.5 (0.5%) | 0.5 (11%) | 210 (3.2%) | 234 (35%) |
| Mangrove | 3315 (2%) | 10,509 (8%) | 0.4 (1.8%) | 1.4 (6%) | 81 (1.3%) | 257 (5%) |
| Seagrass | 92,569 (15%) | 127,720 (43%) | 2.5 (2.3%) | 3.5 (7%) | 762 (9%) | 1,051 (25%) |
| Total | 109,649 (9%) | 153,558 (32%) | 3.5 (1.6%) | 5.5 (7%) | 1053 (5%) | 1542 (11%) |

a: Global estimates based on studies providing data (mean or median values) based on global datasets
b: estimates for Australia. The proportion (maximum and minimum) of Australian ecosystems compared to global estimates is presented in brackets

attempts to promote GHG abatement through the conservation, restoration or creation of VCE[42]. For example, restoring tidal flows to drained coastal areas has been identified as a feasible activity for restoring VCE[42]. Evidence of the effectiveness of VCE restoration for enhancing biomass and soil C sequestration is available[43–45], although Australian case studies are limited[41].

The potential of VCE conservation for climate change mitigation relies on the preservation of their millenary soil C stocks. Following disturbance or conversion of VCE, a portion of the soil C becomes exposed to oxic conditions and decays at a relatively fast rate (estimated at $0.183 \, year^{-1}$), resulting in the remineralization of 85% of the soil C stock exposed to oxic conditions within a decade[46]. Since major historic losses of VCE extent in Australia occurred more than two decades ago, restoration of these areas may have little benefit in terms of avoided GHG emissions.

Assuming a recovery of soil C sequestration after VCE rehabilitation, we estimate that the restoration of an area equivalent to 10% of historic losses of VCE extent in Australia ($5730 \, km^2$) would enhance soil C sequestration by $1.15 \pm 0.91$ Tg $CO_2$-e $year^{-1}$ (mean ± SD; Table 3), reducing annual emissions from land use change in Australia by 6–8%. This can also result in a benefit of US$ $11.5 \pm 9.1$ million per annum (assuming a conservative C trading price of US$10 t $CO_2$-e$^{-1}$, the approximate value paid in Australia's Emissions Reduction Fund auctions). The benefits of restoration vary among VCE and climate regions, averaging $1.4 \pm 1.1$ Mg $CO_2$ ha $year^{-1}$ for tidal marsh (13–16 US$ ha $year^{-1}$), $4.6 \pm 3.3$ Mg $CO_2$ ha $year^{-1}$ for mangroves (15–145 US$ ha $year^{-1}$), and $1.3 \pm 1.2$ Mg $CO_2$ ha $year^{-1}$ for seagrass (10–18 US$ ha $year^{-1}$). These estimates are conservative since potential avoided GHG emissions from soil C following the restoration of historic losses of VCE extent were not accounted for, while the restoration of mangroves would also entail enhanced $CO_2$ sequestration in aboveground biomass.

Targeting blue carbon hotspots for restoration can provide larger benefits per unit area. Mangroves, in particular those occupying tropical regions where there is potential for reversing large, historic losses in coastal wetland extent[42], are hotspots for C sequestration. For example, the restoration of an area equivalent to 10% of historic losses in tropical mangrove extent in Australia ($1150 \, km^2$) would enhance soil C sequestration by $0.65 \pm 0.46$ Tg $CO_2$-e $year^{-1}$, reducing annual emissions from land use change in Australia by 4–5%. It is therefore imperative to develop policies that preserve and restore VCE to mitigate GHG emissions in Australia, where recovery of historic losses offers a vast potential for C sequestration. Importantly, conservation and restoration of VCE also provide enhanced adaptation to climate change through coastal protection and regulation of flooding, as well as biodiversity and fisheries benefits[5,47].

Recent coastal development in Australia continues to result in a net decline in VCE extent, estimated at a minimum 0.03% $year^{-1}$ for Australian mangroves (100–315 ha $year^{-1}$; Table 3)[38]. The decline of tidal marsh extent in Australia remains unknown, but it is likely similar or higher than in mangroves (415–460 ha $year^{-1}$)[14]. Similarly, current decline in seagrass extent in Australia has not been estimated, and here we assume a loss of 0.1% $year^{-1}$ (i.e., 9300–12,800 ha $year^{-1}$) largely resulting from dredging, water quality deterioration and shoreline modification, to estimate potential $CO_2$ emissions resulting from habitat loss[48]. This assumption is conservative relative to global estimates, which are one order of magnitude higher[40]. Assuming that 50% of the aboveground biomass and soil C in the top meter are remineralized after disturbance[4,49], we estimate emissions of 2.1–3.1 Tg $CO_2$-e $year^{-1}$ as a result of current losses of Australian VCE (Table 3), increasing emissions from land use change in Australia by 12–21% per annum. The loss of VCE would also result in a loss of future soil C sequestration of 13–19 Gg $CO_2$-e $year^{-1}$. Thus, conservation actions resulting in avoided losses of VCE in Australia, falling outside business-as-usual scenarios, could result in avoided $CO_2$ emissions and sustained C sequestration valued at 22–31 million US$ per annum (3000–4000 US$ ha $year^{-1}$ for tidal marsh, 2000–22,000 US$ ha $year^{-1}$ for mangroves, or 1500–3000 US$ ha $year^{-1}$ for seagrass).

**Implementation of blue carbon strategies in Australia**. Sustainable management of VCE requires an informed understanding of the ecological and economic significance of changes in natural resources due to threats such as human activities and

**Table 3 Potential annual $CO_2$ emissions from loss of Australian vegetated coastal ecosystems and economic valuation**

**a**

| Ecosystem | Total stock (Soil + Biomass) (Tg C) | Habitat loss per year (ha $year^{-1}$) | C at risk of remineralization (Tg C $year^{-1}$) | Potential $CO_2$ emissions (Tg $CO_2$-e $year^{-1}$) | Economic value of $CO_2$ emissions per year ($10 t $CO_2^{-1}$) ($10^6$ US$) |
|---|---|---|---|---|---|
| Tidal marsh | 212–237 | 413–460 | 0.036–0.040 | 0.13–0.15 | 1.3–1.5 |
| Mangrove | 131–415 | 99–315 | 0.019–0.059 | 0.07–0.22 | 0.7–2.2 |
| Seagrass | 778–1,073 | 9,257–12,772 | 0.53–0.73 | 1.9–2.7 | 19–27 |
| Total | 1121–1725 | 9769–13,547 | 0.58–0.83 | 2.1–3.1 | 21–30 |

**b**

| Ecosystem | Sequestration rates in Australia (Tg C $year^{-1}$) | Habitat loss per year (ha $year^{-1}$) | Lack of C sequestration (Gg C $year^{-1}$) | Potential lack of $CO_2$ sequestration (Gg $CO_2$-e $year^{-1}$) | Economic value of lack of $CO_2$ sequestration per year ($10 t $CO_2^{-1}$) ($10^6$ US$) |
|---|---|---|---|---|---|
| Tidal marsh | 0.48–0.53 | 413–460 | 0.16–0.18 | 0.58–0.65 | 0.0058–0.0065 |
| Mangrove | 0.4–1.4 | 99–315 | 0.13–0.40 | 0.46–1.46 | 0.0046–0.015 |
| Seagrass | 2.5–3.5 | 9,257–12,772 | 3.3–4.6 | 12.2–16.9 | 0.12–0.17 |
| Total | 3.5–5.5 | 9,769–13,547 | 3.6–5.2 | 13.3–19.0 | 0.13–0.19 |

a: Potential gross annual emissions (Tg $CO_2$-e $year^{-1}$) from aboveground biomass and soils as a result of the decline in vegetated coastal ecosystems extent in Australia (0.03% $year^{-1}$ for tidal marshes and mangroves, and 0.1% $year^{-1}$ for seagrasses). Emission estimates assume that 50% of organic carbon (C) stocks in aboveground biomass and in the top meter of soil deposits are remineralized after ecosystem loss (at a rate of 0.183 $year^{-1}$ [46])
b: Potential annual loss of $CO_2$ sequestration capacity in blue carbon soils as a result of current losses in the extent of vegetated coastal ecosystems in Australia, assuming that soil C accretion does not occur after ecosystem loss. Carbon trading price of US$ 10 per ton of $CO_2$. Economic value is expressed in $10^6$ US$

climate change. Current assessments of $CO_2$ emissions from VCE due to land use change in Australia have focused on emissions due to conversion of mangroves and tidal marshes to settlements (Tier 1 estimates), and have not accounted for emissions associated with losses of seagrass nor for conversion of mangrove and tidal marsh to pasture[37]. The Australian Government established the International Partnership for Blue Carbon after the Conference of the Parties to the United Nations Framework on Climate Change conference in Paris 2016. The Australian Government also commissioned a technical review of the inclusion of blue carbon projects in its domestic carbon abatement scheme (the Emissions Reduction Fund), through management of ecosystems towards the enhancement of C storage and/or avoided GHG emissions[42]. Our assessment of Australian national blue carbon storage, accounting for the various climate regions and administrative jurisdictions, provides a basis to estimate potential $CO_2$ abatement through restoration and conservation of VCE.

Our results show that Australia is a hotspot for VCE holding large quantities of blue carbon storage equivalent to 5–11% of blue carbon soil stocks worldwide, despite losses amounting to 47–78% of tidal marsh and mangrove extents, and 20–26% of seagrass extent since European arrival. Therefore, Australia stands to benefit from developing blue carbon-focused climate change mitigation schemes. Restoration of historic losses of VCE together with enhanced conservation of threatened VCE could constitute a mechanism to mitigate Australian $CO_2$ emissions while enhancing ecosystem services and climate adaptation capacity. The estimates reported here provide a pioneer demonstration of the approach required to deliver estimates that can be incorporated into national carbon accounting and underpinning the incorporation of robust blue carbon strategies within Nationally *Determined* Contribution to mitigate climate change. The baseline map of blue carbon in Australia provides an essential underpinning to assess the impact of land use changes and climate change on blue carbon fluxes and stocks. The pioneer assessment at a national, and continental, level reported here provides a methodology beyond the use of Tier 1 approaches currently available in the IPCC Wetland Supplement[50] that provides an exemplar of an approach toward estimating national blue carbon resources elsewhere.

## Methods

**Data acquisition**. Data on C stocks and sequestration rates in Australian tidal marshes, mangrove forests and seagrass meadows were compiled from published data. In addition, unpublished studies from the CSIRO Marine and Coastal Carbon Biogeochemistry Cluster project and other studies by the co-authors were included. The study sites included mono-specific and/or mixed tidal marsh, mangrove and seagrass ecosystems within a variety of depositional environments (from estuarine to exposed coastal areas, and supra-tidal to sub-tidal habitats) across five climate regions (arid, semi-arid, temperate, subtropical and tropical) in Australia. Data from 1553 study sites (593 from tidal marshes, 323 from mangrove forests and 637 from seagrass meadows) on soil C stocks (1103 cores in total), soil C sequestration rates (352 cores in total) and standing C stocks in aboveground biomass (141 measurements in total) were used in this study.

Soil cores were sampled using different coring mechanisms, such as manual percussion and rotation of PVC pipes, vibracoring or using a Russian corer. The cores were sliced at regular intervals, each slice/sample was weighed before and after oven drying to constant weight at 60–70 °C (i.e., dry weight, DW).

The organic C content of the soil organic matter was measured in milled subsamples from multiple slices along cores. The 'Champagne test' was used to determine whether soil samples contained inorganic carbon[51]. The soil core subsamples containing carbonates were acidified with 1 M HCl, centrifuged (3500 RPM; 5 min) and the supernatant with acid residues was removed using a pipette, then washed in deionized water, centrifuged again and the supernatant removed. These residual samples were re-dried (60–70 °C) before C elemental analyses. The method used to remove inorganic C prior to organic C analyses may lead to the loss of part of the organic C (soluble fraction), thereby potentially leading to an underestimation of sediment C content[52]. Where carbonates were absent (all living plant samples and most tidal marsh and mangrove soil samples), bulk soil samples were milled and encapsulated without acid pre-treatment before C analyses. The C content was obtained using a dry combustion elemental analyzer or mass

spectrometer. Percentage soil C on a mass basis was calculated for the bulk (pre-acidified) samples.

Data on soil accumulation rates from 315 cores derived by means of $^{210}$Pb (last century) and/or radiocarbon (last millennia) was compiled. Concentration profiles of $^{210}$Pb along the upper part of the sediment cores were determined by alpha spectrometry through the measurement of $^{210}$Po using Passivated Implanted Planar Silicon (PIPS) detectors (CANBERRA, Mod. PD-450.18 A.M) after acid digestion of the samples, assuming radioactive equilibrium between the $^{210}$Pb and $^{210}$Po radionuclides. After alpha spectrometry, selected samples from each core were analyzed for $^{226}$Ra by ultra-low background liquid scintillation counting (LSC, Quantulus 1220) or gamma spectrometry through the emission lines of its daughter radionuclides $^{214}$Pb and $^{214}$Bi (295.2, 351.9, and 609.3 KeV). The concentration profiles of excess $^{210}$Pb were determined by subtraction of $^{226}$Ra from total $^{210}$Pb concentrations along each core. Gamma spectrometry measurements were conducted in some cores using semi-planar intrinsic germanium high purity coaxial detectors with 40% efficiency, housed in a lead shield, coupled to a multichannel analyzer. $^{210}$Pb activity was determined by the direct measurement of 46.5 KeV gamma peak. Sediment accumulation rates were obtained by applying the Constant Rate of Supply (CRS[53]) or the Constant Flux: Constant Sedimentation models (CF:CS[54]). The $^{239+240}$Pu activities were measured in a sector ICPMS[55].

Samples of bulk soil, plant debris and shells along the cores were radiocarbon dated following standard procedures[56]. The $^{14}$C dates from seagrass cores were calibrated using the marine13 calibration curve[57] considering a local Delta R ranging from 3 to 71 years as a function of study site[58]. The corrected ages were used to produce an age-depth model (linear regression) to estimate sediment accumulation rates.

**Data analyses**. To allow direct comparison among study sites, the C storage per unit area (cumulative stocks, mass C m$^{-2}$) was standardized to 1 m-thick deposits (i.e., length of the soil cores sampled). When necessary, we inferred C stocks below the limits of the reported data to 1 m, extrapolating linearly integrated values of C content (cumulative C stock per unit area) with depth. Correlation between extrapolated C stocks and measured C stocks in sediment cores ≥1 m was $r^2 = 0.87$ ($P < 0.001$). The C sequestration rates (mass C m$^{-2}$ year$^{-1}$) were calculated by multiplying average C concentration by the sediment accumulation rate (mass m$^{-2}$ year$^{-1}$) in each core (where quantified). Estimates of aboveground biomass per unit area were obtained by drying and weighing aboveground materials for tidal marshes and seagrasses, and using field measurements and allometric equations (specific to the region and species) for mangroves[59,60].

All analyses were performed using Generalized Linear Model procedures in SPSS v. 14.0. A Generalized Linear Model was used to consider the potential non-independence of samples taken within habitats. All response variables (C stocks in aboveground biomass and soil C stocks and sequestration rates) were square-root transformed prior to analyses and had homogenous variances. Climate region (arid, semi-arid, temperate, subtropical, and tropical) and ecosystem type (tidal marsh, mangrove and seagrass) were treated as fixed factors in all statistical models (probability distribution: normal; link function: identity).

Potential C stock losses (mass C) and $CO_2$ emissions (mass $CO_2$-e year$^{-1}$) were estimated based on 0.03% annual ecosystem area loss for tidal marshes and mangroves, and 0.1% year$^{-1}$ for seagrass, and accounted for the sum of C stocks in aboveground biomass and the top meter of soils, assuming that 50% of total C stocks are lost and remineralized into $CO_2$ after disturbance[4,49].

The upscaling of each habitat polygon was performed by multiplying the average ± SD soil C stocks, sequestration rates, and standing C stocks in the aboveground biomass for each ecosystem within each climate region by the specific ecosystem area to obtain blue carbon estimates at climate region scale (arid, semi-arid, temperate, subtropical and tropical, adapted from[61]; Fig. 1) and administrative jurisdictions within Australia (Northern Territory, Queensland, New South Wales, Victoria, Tasmania, South Australia and Western Australia). The datasets on biomass C stocks ($N = 37$ for mangroves and $N = 52$ for both tidal marshes and seagrasses) and on soil C sequestration rates for mangroves ($N = 24$) and seagrasses ($N = 36$) were limited, which resulted in data gaps within climate regions (Supplementary Fig. 1 and Supplementary Table 3). For example, estimates of biomass C stocks in tidal marsh are lacking for arid and tropical regions (Supplementary Table 3). In order to estimate C storage in VCE around Australia, C data from the nearest climate region was used when data was not available. The extents of each ecosystem considered to scale up Australia-wide estimates of C stocks and sequestration (based on climate regions) are presented in Table 2 and Supplementary Table 2.

Substantial data gaps for blue carbon stocks and sequestration rates exists in parts of the country (Supplementary Fig. 1 and Supplementary Table 3). Most notably, there are limited data over much of northern Australia, where ecosystem extent is greatest for all three VCE. Stocks, sequestration rates, and losses of C are also poorly quantified in converted or modified systems and there are few studies of C sequestration capacity in restored ecosystems[49]. The robustness of the global assessments presented here relies on a number of estimates, including the extent of Australian VCE, annual loss rates, degree and fate of soil C loss after disturbance, differences in the impact of different types of disturbances or management activities, and the type of C trading methodology used. Therefore, the potential C

abatement and its economic value may vary across VCE, and with management and political scenarios.

## Data availability

A Source Data File, containing the raw data underlying the research and all figures and tables presented in our paper, is available in the Supplementary Information. The spatial datasets that support the findings of this study have been deposited in the Commonwealth Scientific and Industrial Research Organisation portal with the identifier https://doi.org/10.25919/5d3a8acc9b598.

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

## Acknowledgements

This project was supported by the CSIRO Marine and Coastal Carbon Biogeochemical Cluster, CSIRO Oceans and Atmosphere, the ECU Faculty Research Grant Scheme and Early Career Research Grant Schemes, UTS Plant Functional Biology and Climate Change Cluster, NSW Southeast Local Land Services, Department of Environment, Land, Water and Planning (DELWP), Parks Victoria, Victorian Coastal Catchment Management Authorities (GHCMA, CCMA, PPWCMA, WGCMA, EGCMA), University of Queensland Centennial Scholarship, Hodgkin Trust Scholarship, Australian Institute of Nuclear Science and Engineering, Northern Territory Government Innovation Grant, Australian Research Council (DE130101084, DE140101733, DE150100581, DE160100443, DE170101524, DP150103286, DP150102092, DP160100248, DP160100248, DP180101285, LE140100083, LE170100219, LP150100519, LP160100242 and LP110200975), the Generalitat de Catalunya (MERS 2014 SGR-1356), the ICTA 'Unit of Excellence' (MinECo, MDM2015-0552), Obra Social "LaCaixa", SUMILEN, CTM 2013-47728-R, Ministry of Economy and Competitiveness and UKM-DIP-2017-005. The authors are grateful to G. Bastyan, D. Kyrwood, G. Davis, J. Bongiovanni, A. Jesse, Q. Hua, A. Zawadzki, J. Gudiño, P. Bray, H. Markham, M. Lepore, K-le Gómez-Cabrera, and J. Pandolfi for their help in field and/or laboratory tasks.

## Author contributions

C.M.D. and C.E.L. conceived and designed the research, O.S. coordinated data acquisition with contributions from all co-authors, analyzed the data, and drafted the first version of the paper. O.S., C.E.L., T.A., P.I.M., R.C., S.P., A.A.-O., L.B., J.B., C.B., P.C., R.M.C., P.D., A.E., C.J.E.L., B.D.E., M.A.H., P.H., L.B.H., C.R.J.K., J.J.K., G.A.K., K.K., A.L., S.Y.L., P.S.L., D.T.M., N.M., P.M., M.A.M., R.M., P.J.R., C.R., M.R., R.R., C.S., J.S.-V., J.S., C.S., I.S., C.S., A.D.L.S., T.C., S. M.T.-T. and C.M.D. contributed by providing their data and improving the paper.

## Additional information

**Competing interests:** The authors declare no competing interests.

Oscar Serrano[1], Catherine E. Lovelock[2,3], Trisha B. Atwood[3,4], Peter I. Macreadie[5], Robert Canto[3,6], Stuart Phinn[3,6], Ariane Arias-Ortiz[7], Le Bai[8], Jeff Baldock[9], Camila Bedulli[10,11], Paul Carnell[5], Rod M. Connolly[12], Paul Donaldson[13], Alba Esteban[1], Carolyn J. Ewers Lewis[5], Bradley D. Eyre[14], Matthew A. Hayes[2,3,12], Pierre Horwitz[1], Lindsay B. Hutley[8], Christopher R.J. Kavazos[1,15], Jeffrey J. Kelleway[16], Gary A. Kendrick[10,17], Kieryn Kilminster[17,18], Anna Lafratta[1], Shing Lee[12,19], Paul S. Lavery[1,20], Damien T. Maher[14], Núria Marbà[21], Pere Masque[1,7,10,22], Miguel A. Mateo[1,20], Richard Mount[23], Peter J. Ralph[24], Chris Roelfsema[6], Mohammad Rozaimi[1,25], Radhiyah Ruhon[10,26], Cristian Salinas[1,27], Jimena Samper-Villarreal[3,28,29], Jonathan Sanderman[9,30], Christian J. Sanders[31], Isaac Santos[31], Chris Sharples[23], Andrew D.L. Steven[32], Toni Cannard[32], Stacey M. Trevathan-Tackett[5] & Carlos M. Duarte[10,33]

[1]School of Science and Centre for Marine Ecosystems Research, Edith Cowan University, Joondalup, WA 6027, Australia. [2]School of Biological Sciences, University of Queensland, St. Lucia, QLD 4072, Australia. [3]The Global Change Institute, University of Queensland, St. Lucia, QLD 4072, Australia. [4]Department of Watershed Sciences and Ecology Center, Utah State University, Logan, UT 84322, USA. [5]Centre for Integrative Ecology, School of Life and Environmental Sciences, Deakin University, Geelong, Burwood Campus, Geelong, VIC 3125, Australia. [6]Remote Sensing Research Centre/Joint Remote Sensing Research Program, School of Earth and Environmental Sciences, University of Queensland, Queensland, QLD 4072, Australia. [7]Institut de Ciència i Tecnologia Ambientals and Departament de Física, Universitat Autònoma de Barcelona, 08193 Bellaterra, Spain. [8]Research Institute for the Environment and Livelihoods, Charles Darwin University, Casuarina, NT 0810, Australia. [9]CSIRO Agriculture and Food, Locked Bag 2, Glen Osmond, SA 5064, Australia. [10]UWA Oceans Institute, The University of Western Australia, Crawley, WA 6009, Australia. [11]Instituto de Biociências de Botucatu, Universidade Estadual Paulista, Botucatu 18618-970, Brazil. [12]Australian Rivers Institute—Coast and Estuaries, School of Environment andScience, Griffith University, Gold Coast, QLD 4222, Australia. [13]BMT Environment, Newcastle, NSW 2292, Australia. [14]Centre for Coastal Biogeochemistry, School of Environment, Science and Engineering, Southern Cross University, Lismore, NSW 2480, Australia. [15]School of Biological, Earth and Environmental Sciences, University of New South Wales, Kensington, NSW 2052, Australia. [16]School of Earth, Atmospheric and Life Sciences, University of Wollongong, Wollongong, NSW 2522, Australia. [17]School of Biological Sciences, The University

of Western Australia, Crawley, WA 6009, Australia. [18]Department of Water and Environmental Regulation, Locked Bag 10, Joondalup DC, WA 6027, Australia. [19]Simon FS Li Marine Science Laboratory, Chinese University of Hong Kong, Shatin, Hong Kong. [20]Centre d'Estudis Avançats de Blanes-CSIC, 17300 Blanes, Spain. [21]Global Change Research Group, IMEDEA (CSIC-UIB), Institut Mediterrani d'Estudis Avançats, Miquel Marquès 21, 07190 Esporles, Spain. [22]School of Physics, The University of Western Australia, 35 Stirling Highway, Crawley, WA 6009, Australia. [23]Discipline of Geography and Spatial Sciences, School of Technology, Environments and Design, University of Tasmania, Hobart, TAS 7001, Australia. [24]Climate Change Cluster, University of Technology Sydney, PO Box 123, Broadway, NSW 2007, Australia. [25]Centre for Earth Sciences and Environment, Faculty of Science and Technology, Universiti Kebangsaan Malaysia, 43600 Bangi, Selangor, Malaysia. [26]Faculty of Marine Science and Fisheries, Hasanuddin University, Jl. Perintis Kemerdekaan Km.10, Tamalanrea, Makassar 90245, Indonesia. [27]Marine and Coastal Research Institute "José Benito Vives De Andréis" INVEMAR, Calle 25 No. 2-55, Santa Marta, Colombia. [28]Centro de Investigación en Ciencias del Mar y Limnología (CIMAR), Ciudad de la Investigación, Universidad de Costa Rica, San Pedro, San José 11501-2060, Costa Rica. [29]Marine Spatial Ecology Lab, University of Queensland, St Lucia, QLD 4072, Australia. [30]Woods Hole Research Center, Falmouth, MA 02540, USA. [31]National Marine Science Centre, Southern Cross University, PO Box 4321, Coffs Harbour, NSW 2450, Australia. [32]CSIRO Oceans and Atmosphere, Queensland Biosciences Precinct, 306 Carmody Rd, St. Lucia, QLD 4067, Australia. [33]Red Sea Research Center (RSRC) and Computational Bioscience Research Center (CBRC), King Abdullah University of Science and Technology (KAUST), Thuwal 23955-6900, Saudi Arabia

