## [Peer Review File · Nature Communications]

Reviewers' comments:

Reviewer #1 (Remarks to the Author):

Study overview:

This study provides a comprehensive national assessment of blue carbon storage and potential annual CO₂ emissions in vegetated coastal ecosystems (VCE) in Australia. C stocks in biomass and soils, and CO₂ sequestration rates in VCE are presented by Australian administrative jurisdictions and explained on the basis ecosystem type and climate zones. C stocks and sequestration rates estimates used in this study were compiled from published studies and from unpublished data from co-authors of this study. Total and areal estimates were computed by multiplying the average values for each ecosystem within each climate region by the specific ecosystem area. These baseline values were then used to estimate ecological and economic restoration benefits (eg, offsetting CO₂ emissions, reducing land use change) using hypothetical land use change scenarios. This study shows the potential for VCE conservation and restoration to mitigate GHG emissions in Australia and support policies that contribute to climate change mitigation at the national level.

Journal's specific questions:

What are the major claims of the paper?

The major claim of this paper is providing the most comprehensive blue carbon stocks and CO₂ sequestration rates inventory, and potential annual CO₂ emissions for any nation to-date. The authors claim to provide a methodology beyond the use of IPCC Tier 1 and to provide a methodological framework to estimate national blue carbon resources elsewhere.

Are they novel and will they be of interest to others in the community and the wider field? If the conclusions are not original, it would be helpful if you could provide relevant references.

This study synthesizes data from published and unpublished sources for VCE in Australia and will certainly be of interest to others in the community. The results are original and provide an important baseline for creation or revision of robust blue carbon strategies in Australia and for assessing the impact of land use change and climate change on blue carbon fluxes and stocks. The approach used in this study is also useful for other nations to develop or review their blue carbon and potential CO_{2e} emissions inventories.

Is the work convincing, and if not, what further evidence would be required to strengthen the conclusions?

I find this work very straightforward and grounded on a comprehensive and systematic review of the existing literature. I have a moderate concern regarding part of the soil cores used in their analyses. Some soil cores included in their estimates apparently have not been treated for carbonate (inorganic carbon) removal. Thus, total carbon (that is, organic + inorganic) should be reported instead of total organic carbon. The problem here is that these non-treated soil samples could yield inflated soil organic carbon stock estimates. I believe there are ways to work around this issue and am suggesting authors to either review some of their statements (eg, report total carbon instead) or provide evidence that the inclusion of such cores does not influence soil organic carbon estimates. Please see 'other comments' for more details.

On a more subjective note, do you feel that the paper will influence thinking in the field?

While this paper will be without a question a valuable contribution to the field, I do not feel it will influence the thinking in the field. Yet comprehensive this study is mostly descriptive. It neither challenges or advances any theory in the field nor it offers an alternative hypothesis to our current understanding about processes that control C storage and sequestration rates in blue carbon ecosystems.

We would also be grateful if you could comment on the appropriateness and validity of any statistical analysis, as well the ability of a researcher to reproduce the work, given the level of detail provided.

The statistical analyses used in the manuscript is clearly described, appropriately applied and can be easily reproduced by others.

Other comments:

Ln 115: I would recommend adding “, particularly for tidal marshes and seagrasses” after “are lacking”, as blue carbon stocks in mangroves have been reasonably mapped at national and continental scales, including recent publications by co-authors of this manuscript.

Ln 132: Biomass means living mass. Otherwise it's necromass. It seems redundant to use living to denote biomass.

Ln 170-172: It would be helpful to provide mean areal estimates for these temperate and tropical/subtropical species.

Ln 187-181: Could higher C stocks in biomass and soil in seagrasses inhabiting arid climate regions be partly explained by absent or ephemeral riverine input in these regions, thus decreasing water turbidity? On the contrary, tropical estuarine environments are generally subjected to higher run-off and higher total suspended solids, which could in turn decrease seagrass above and belowground productivity by limiting PAR. I think incorporating such coastal processes into your discussion could be helpful to explain some of the spatial variability you found across climate regions, in addition to the biological explanation you provided.

Ln 194-195, 197: It would be helpful to provide mean areal estimates for these tropical vs. arid and temperate species.

Ln 224-225: Where these loss rates come from? Please provide the source.

Ln 242: Please double-check these values. I can follow that CO₂ sequestration capacity of Australian VCE equates to 70% of CO₂ emissions from land use change but not to 140% from the ranges provided.

Ln 371-373: I am skeptical about relying on visual inspection to determine whether or not a soil sample has carbonates in its composition. This is a red flag in your analyses and should be reported properly. C stock estimates resulting from non-acidified cores should be reported as total carbon (that is, organic + inorganic), not total organic carbon. Could you work around this issue by providing some evidence that C stock estimates resulting from non-acidified cores are similar to acid-treated ones for some choice sites? Or could you find support in the literature to show that some of these sites from where non-acidified cores were retrieved are carbonate free? I suggest you to somehow account for potential overestimates resulting from the inclusion of non-treated samples for carbonate removal prior TOC analysis.

Ln 392-395: C sequestration potential is key in this study, and it is suggested to vary across climate zones. However, you mention here that differences in carbon accumulation rates among climate regions may arise from differences in methods used for estimating soil accumulation rates. This should be presented in the main text as it may influence the interpretation of some of your results.

Fig. 1: It is not clear what stacked bars represent in Fig.1B. Please clarify in the caption or add a legend.

Fig. 2: Fig. 2 does not quite show Australian climate zones. For clarity I would suggest combining either Fig. 1 or Fig. 2 with Supplementary Fig. 1 into a redesigned Fig. 1 or 2.

Table 3a: The totals in this table are noted as 'This study' when in fact these are just the sum published values.

Reviewer #2 (Remarks to the Author):

In this paper, the authors provide a robust and extensive summary of blue C ecosystems in Australia. They synthesize aboveground and belowground C stocks and translate that data into sequestration and emissions. The ultimate goal of doing this is to convey the importance that VCE play in C storage and can be used as a mitigation strategy in the fight against climate change.

This paper is novel in that the authors provide a NATIONAL-level compilation and synthesis of blue carbon data. While researchers in other countries are currently in the process of doing this (i.e., Coastal Carbon Research Coordination Network, North America), this is the first synthesis of this type of data to be published at this scale and with a specific audience target of policymakers. This paper is timely and important, most of the methods are sound, and the data and arguments are compelling, but I do have a specific concern about how the data was scaled, particularly for tidal marshes.

Figure 2 presents scaled up estimated of C storage in VCE in Australia and is a vital result for the main purpose of this paper. Fig 2b shows aboveground C biomass over the entire continent of Australia, but supp fig 2b., which shows the actual data measured to create this scaled up version, shows that aboveground C biomass was only sampled in 2 locations, both temperate. I do not see how you can scale across ecosystems types and geographical range with this little data. There are no data points from the northern coast, but Fig 2 shows scaled up data for that region, not only for aboveground biomass but also soil c stocks and C seq. rate. L 435-437 mentions data gaps, especially for the northern region, but this does not address how you dealt with these data gaps. L 426-427 states "These estimates were based on the nearest region when data was not available" So based on this, you are scaling up from two, temperate sites to the entire country, which represents 4 different ecosystem types? This does not seem appropriate, but please correct me if I am interpreting this

incorrectly. Here, I would like to see either a more thorough explanation of how the scaling was done (perhaps more data was used than shown in fig. 2) or more data incorporated into the scaling exercise, especially from regions currently underrepresented in the data

Other minor comments:

L165-168 – Based on the table cited, this statement is not true. In Supp table 1, aboveground biomass is not significantly different among climate regions ($p=0.535$), but soil C stocks (0.016) and seq. rates (<0.01) are. You state that none of these are sig diff.

L165 – 168 – Also, you preface this statement by stating this statement only applies to tidal marshes, but nowhere in supp table 1 does it state what VCE type that these analyses are for.

L168-169 – Here you state that aboveground biomass is higher in temperate marshes compared to other ecosystem types, but supp fig 2 only shows data from two points, both in temperate marshes. How did you do this analysis to make this conclusion (see above note on this for more context)

Fig. 1b – Why are there two shades of color in the bar graph? This is not explained anywhere.

Fig. 2 – Make sure your colors are color blind friendly. In this figure, red and green together would be indistinguishable to some.

Response to reviewers

Reviewer #1 (Remarks to the Author):

Study overview:

This study provides a comprehensive national assessment of blue carbon storage and potential annual CO₂ emissions in vegetated coastal ecosystems (VCE) in Australia. C stocks in biomass and soils, and CO₂ sequestration rates in VCE are presented by Australian administrative jurisdictions and explained on the basis ecosystem type and climate zones. C stocks and sequestration rates estimates used in this study were compiled from published studies and from unpublished data from co-authors of this study. Total and areal estimates were computed by multiplying the average values for each ecosystem within each climate region by the specific ecosystem area. These baseline values were then used to estimate ecological and economic restoration benefits (eg, offsetting CO₂ emissions, reducing land use change) using hypothetical land use change scenarios. This study shows the potential for VCE conservation and restoration to mitigate GHG emissions in Australia and support policies that contribute to climate change mitigation at the national level.

Thank you very much for reviewing our work. We have addressed all your comments and concerns in this revised version of the manuscript (see below).

Journal's specific questions:

What are the major claims of the paper?

The major claim of this paper is providing the most comprehensive blue carbon stocks and CO₂ sequestration rates inventory, and potential annual CO₂ emissions for any nation to-date. The authors claim to provide a methodology beyond the use of IPCC Tier 1 and to provide a methodological framework to estimate national blue carbon resources elsewhere.

Are they novel and will they be of interest to others in the community and the wider field?

If the conclusions are not original, it would be helpful if you could provide relevant references.

This study synthesizes data from published and unpublished sources for VCE in Australia and will certainly be of interest to others in the community. The results are original and provide an important baseline for creation or revision of robust blue carbon strategies in Australia and for assessing the impact of land use change and climate change on blue carbon fluxes and stocks. The approach used in this study is also useful for other nations to develop or review their blue carbon and potential CO₂e emissions inventories.

Thank you

Is the work convincing, and if not, what further evidence would be required to strengthen the conclusions?

I find this work very straightforward and grounded on a comprehensive and systematic review of the existing literature. I have a moderate concern regarding part of the soil cores used in their analyses. Some soil cores included in their estimates apparently have not been

treated for carbonate (inorganic carbon) removal. Thus, total carbon (that is, organic + inorganic) should be reported instead of total organic carbon. The problem here is that these non-treated soil samples could yield inflated soil organic carbon stock estimates. I believe there are ways to work around this issue and am suggesting authors to either review some of their statements (eg, report total carbon instead) or provide evidence that the inclusion of such cores does not influence soil organic carbon estimates. Please see 'other comments' for more details.

All soil samples containing carbonates were treated for inorganic carbon removal, as explained in the Methods section. In particular, carbonates were absent in living plant samples, and in most tidal marsh and mangrove soil samples. The 'Champagne test' (Burt, 2014) was used to determine whether tidal marsh and mangrove soil samples contained inorganic carbon. This was clarified in the methods section:

New text reads (L370-372): "The 'Champagne test' was used to determine whether soil samples contained inorganic carbon⁵². The soil core sub-samples containing carbonates were acidified with 1 M HCl, ..."

New reference ⁵²: Burt, R. Soil Survey Staff. Soil Survey Laboratory Methods Manual. Soil Survey Investigations Report 42, Version 5.0. US Department of Agriculture, Natural Resources Conservation Service, National Soil Survey Center (2014).

On a more subjective note, do you feel that the paper will influence thinking in the field?

While this paper will be without a question a valuable contribution to the field, I do not feel it will influence the thinking in the field. Yet comprehensive this study is mostly descriptive. It neither challenges or advances any theory in the field nor it offers an alternative hypothesis to our current understanding about processes that control C storage and sequestration rates in blue carbon ecosystems.

We would also be grateful if you could comment on the appropriateness and validity of any statistical analysis, as well the ability of a researcher to reproduce the work, given the level of detail provided.

The statistical analyses used in the manuscript is clearly described, appropriately applied and can be easily reproduced by others.

Thank you

Other comments:

Ln 115: I would recommend adding " , particularly for tidal marshes and seagrasses" after "are lacking", as blue carbon stocks in mangroves have been reasonably mapped at national and continental scales, including recent publications by co-authors of this manuscript.

Added as suggested.

Text now reads (L112-115): "However, this requires strong scientific evidence, and whereas reports of C stocks and sequestration rates in VCE have recently increased exponentially^{5,9-11}, comprehensive estimates of blue carbon storage at national and continental scales are lacking, particularly for tidal marshes and seagrass."

Ln 132: Biomass means living mass. Otherwise it's necromass. It seems redundant to use living to denote biomass.

The term "biomass" refers to materials derived from living organisms, but it can be used to describe living and/or dead materials in the blue carbon literature. For example, a timber table is basically plant biomass, even though it is dead. Therefore, we preferred to leave the term 'living biomass' in the manuscript to clarify that our estimates do not include the necromass.

Ln 170-172: It would be helpful to provide mean areal estimates for these temperate and tropical/subtropical species.

The available maps of tidal marsh extent in Australia do not differentiate among species, only data on the total area in each climate region is available (see Supplementary Table 2). Therefore, it was not possible to include this information.

Ln 187-181: Could higher C stocks in biomass and soil in seagrasses inhabiting arid climate regions be partly explained by absent or ephemeral riverine input in these regions, thus decreasing water turbidity? On the contrary, tropical estuarine environments are generally subjected to higher run-off and higher total suspended solids, which could in turn decrease seagrass above and belowground productivity by limiting PAR. I think incorporating such coastal processes into your discussion could be helpful to explain some of the spatial variability you found across climate regions, in addition to the biological explanation you provided.

Thank you. This is a plausible hypothesis that has been included in the manuscript as suggested.

New text reads (L201-204): "In addition, the scarcity of riverine inputs in arid regions likely results in higher irradiance reaching the seafloor compared to other climate regions with higher run-off, thereby enhancing above- and belowground seagrass productivity^{13,23} and soil C stocks in arid regions."

Ln 194-195, 197: It would be helpful to provide mean areal estimates for these tropical vs. arid and temperate species.

As for tidal marshes and mangroves, the available maps of seagrass extent in Australia do not differentiate among species, only data on the total area in each climate region is available (see Supplementary Table 2). Therefore, it was not possible to include this information.

Ln 224-225: Where these loss rates come from? Please provide the source.

Thanks for noting that references were missing here. The references sustaining the loss rates provided in our manuscript were listed a few lines below. In order to avoid repetition, we simplified the statement in the previous lines 224-225.

Previous text read: "Nevertheless, the extent of VCE in Australia makes it a global blue carbon hotspot and the 47–78% loss of tidal marsh and mangrove extent, and 20–26% loss of seagrasses since European arrival, ..."

New text reads (L229-232): “Nevertheless, the extent of VCE in Australia makes it a global blue carbon hotspot and the loss of VCE extent since European arrival, provides unique opportunities for Australia to mitigate emissions through blue carbon strategies.”

Detailed and referenced information of loss was provided below.

Text read (L253-256): “Historic losses of VCE extent after European settlement in the 19th century in Australia have been estimated at 13,800 km² for tidal marsh¹⁴ (47–50% loss of original extent), 11,500 km² for mangroves^{33,39} (52–78% loss of original extent) and 32,000 km² for seagrass⁴⁰ (20–26% loss of original extent).”

Ln 242: Please double-check these values. I can follow that CO₂ sequestration capacity of Australian VCE equates to 70% of CO₂ emissions from land use change but not to 140% from the ranges provided.

The 70-140% range estimated resulted from comparing the minimum value of the range of soil CO₂ sequestration of Australian VCE (13 Tg CO₂-e yr⁻¹) with the maximum value of the range of CO₂ emissions from land use change in Australia (18.4 Tg CO₂-e yr⁻¹); and by comparing the maximum estimate of soil CO₂ sequestration of Australian VCE (20 Tg CO₂-e yr⁻¹) with the minimum estimate of CO₂ emissions from land use change in Australia (14.4 Tg CO₂-e yr⁻¹). We double checked the values and they are correct.

Ln 371-373: I am skeptical about relying on visual inspection to determine whether or not a soil sample has carbonates in its composition. This is a red flag in your analyses and should be reported properly. C stock estimates resulting from non-acidified cores should be reported as total carbon (that is, organic + inorganic), not total organic carbon. Could you work around this issue by providing some evidence that C stock estimates resulting from non-acidified cores are similar to acid-treated ones for some choice sites? Or could you find support in the literature to show that some of these sites from where non-acidified cores were retrieved are carbonate free? I suggest you to somehow account for potential overestimates resulting from the inclusion of non-treated samples for carbonate removal prior TOC analysis.

The ‘Champagne test’, a standard protocol in soil analysis (Burt, 2014) was used to determine whether tidal marsh and mangrove soil samples contained inorganic carbon. All seagrass soil samples, and all tidal marsh and mangrove soil samples that contained carbonates, were acidified prior to organic carbon analysis. This was clarified in the methods section:

New text reads (L370-372): “The ‘Champagne test’ was used to determine whether soil samples contained inorganic carbon⁵². The soil core sub-samples containing carbonates were acidified with 1 M HCl, ...”

Ln 392-395: C sequestration potential is key in this study, and it is suggested to vary across climate zones. However, you mention here that differences in carbon accumulation rates among climate regions may arise from differences in methods used for estimating soil accumulation rates. This should be presented in the main text as it may influence the interpretation of some of your results.

This text has been moved into the main text as suggested (L226-229).

Fig. 1: It is not clear what stacked bars represent in Fig.1B. Please clarify in the caption or add a legend.

Thanks for noting this. The stacked bars represent the maximum and minimum estimates. This has been indicated in the caption as suggested.

New text added (L663-664): “The stacked bars represent the maximum and minimum estimates.”

Fig. 2: Fig. 2 does not quite show Australian climate zones. For clarity I would suggest combining either Fig. 1 or Fig. 2 with Supplementary Fig. 1 into a redesigned Fig. 1 or 2.

The Supplementary Fig. 1 has been merged into Fig. 1 as suggested (pages 27-28).

Table 3a: The totals in this table are noted as ‘This study’ when in fact these are just the sum published values.

We removed the notation to ‘This study’ from the table as suggested (now Table 2 in page 32).

Reviewer #2 (Remarks to the Author):

In this paper, the authors provide a robust and extensive summary of blue C ecosystems in Australia. They synthesize aboveground and belowground C stocks and translate that data into sequestration and emissions. The ultimate goal of doing this is to convey the importance that VCE play in C storage and can be used as a mitigation strategy in the fight against climate change.

This paper is novel in that the authors provide a NATIONAL-level compilation and synthesis of blue carbon data. While researchers in other countries are currently in the process of doing this (i.e., Coastal Carbon Research Coordination Network, North America), this is the first synthesis of this type of data to be published at this scale and with a specific audience target of policymakers. This paper is timely and important, most of the methods are sound, and the data and arguments are compelling, but I do have a specific concern about how the data was scaled, particularly for tidal marshes.

Thank you very much for reviewing our work. We have addressed all your comments and concerns in this revised version of the manuscript (see below).

Figure 2 presents scaled up estimated of C storage in VCE in Australia and is a vital result for the main purpose of this paper. Fig 2b shows aboveground C biomass over the entire continent of Australia, but supp fig 2b., which shows the actual data measured to create this scaled up version, shows that aboveground C biomass was only sampled in 2 locations, both temperate. I do not see how you can scale across ecosystems types and geographical range with this little data. There are no data points from the northern coast, but Fig 2 shows scaled up data for that region, not only for aboveground biomass but also soil c stocks and C seq. rate. L 435-437 mentions data gaps, especially for the northern region, but this does not address how you dealt with these data gaps. L 426-427 states “These estimates were based on

the nearest region when data was not available” So based on this, you are scaling up from two, temperate sites to the entire country, which represents 4 different ecosystem types? This does not seem appropriate, but please correct me if I am interpreting this incorrectly. Here, I would like to see either a more thorough explanation of how the scaling was done (perhaps more data was used than shown in fig. 2) or more data incorporated into the scaling exercise, especially from regions currently underrepresented in the data

Our estimates of nation-wide biomass C stocks, and soil C stocks and accumulation rates in VCE have limitations that were listed in L444-454. The Supplementary Fig. 2 (now Supplementary Figure 1) can be confusing because the dots overlap. However, detailed data descriptors (e.g. N, mean, median, SD) were provided in Table 1 for Australian-wide estimates, and in Supplementary Table 3 for estimates within climate regions in Australia. Our previous dataset on tidal marsh biomass C stocks had 9 values. We collected new data (now N = 52) that strengthened the dataset, following the methods outlines in L412-414. Yet, gaps still exist for tropical and arid regions.

We detailed these limitations in the manuscript, and provided further explanations on how the scaling up was run.

New text reads (L435-441): “The datasets on biomass C stocks (N = 37 for mangroves and N = 52 for both tidal marshes and seagrasses) and on soil C sequestration rates for mangroves (N=24) and seagrasses (N=36) were limited, which resulted in data gaps within climate regions (Supplementary Fig. 1 and Supplementary Table 3). For example, estimates of biomass C stocks in tidal marsh are lacking for arid and tropical regions (Supplementary Table 3). In order to estimate C storage in VCE around Australia, C data from the nearest climate region was used when data was not available.”

Other minor comments:

L165-168 – Based on the table cited, this statement is not true. In Supp table 1, aboveground biomass is not significantly different among climate regions ($p=0.535$), but soil C stocks (0.016) and seq. rates (<0.01) are. You state that none of these are sig diff.

L165 – 168 – Also, you preface this statement by stating this statement only applies to tidal marshes, but nowhere in supp table 1 does it state what VCE type that these analyses are for.

L168-169 – Here you state that aboveground biomass is higher in temperate marshes compared to other ecosystem types, but supp fig 2 only shows data from two points, both in temperate marshes. How did you do this analysis to make this conclusion (see above note on this for more context)

After including the new data on tidal marsh biomass C stocks, we found significant differences among climatic regions ($P = 0.023$). The Supplementary Table 1 shows the results of the General Linear models (main effects and post-hoc tests). The main effects are presented in the main body of the table, and the results of post-hoc tests (i.e. significant interactions only) are indicated with numbers in superscript.

New text reads (L165-169): “For tidal marshes, the C stocks in aboveground biomass per unit area were up to 6-fold higher in temperate regions compared to semi-arid and subtropical

regions ($P < 0.05$), while soil C stocks and sequestration rates per unit area were not significantly different among climate regions ($P > 0.05$; Supplementary Table 1).”

We indicated the categories of both climate region and ecosystem factors in the caption of Supplementary Table 1.

New caption reads: “Supplementary Table 1. General Linear Models. Living aboveground biomass organic carbon (C) stock, soil C stock (in the top meter) and soil C sequestration rates in response to climate region and ecosystem type (fixed effects), and interaction between climate region and ecosystem type. Climate region: arid, semi-arid, temperate, subtropical and tropical. Ecosystem: tidal marsh, mangrove, seagrass. Significant interactions from post-hoc HSD tests are indicated with numbers in superscript ($P < 0.05$).”

Fig. 1b – Why are there two shades of color in the bar graph? This is not explained anywhere. Thanks for noting this. The stacked bars represent the maximum and minimum estimates. This has been indicated in the caption as suggested.

New caption Figure 1C reads: “C) Organic carbon stocks in living aboveground biomass and soils (in the top meter), and C sequestration rates per unit area (Mg C ha^{-1}) and across Australia (Tg C). The stacked bars represent the maximum and minimum estimates.”

Fig. 2 – Make sure your colors are color blind friendly. In this figure, red and green together would be indistinguishable to some.

Thanks for noting this. We made the figures comprehensible for colour-blind readers (Fig. 1 and 2, and Supp. Fig. 1).

REVIEWERS' COMMENTS:

Reviewer #1 (Remarks to the Author):

Thank you for clarifying the points I raised.

Reviewer #2 (Remarks to the Author):

The authors have addressed all of my comments adequately. This is now a strong paper for this field